# The Effect of Pregnancy on Dermatological Disorders: A Systematic Review

**DOI:** 10.3390/clinpract15040068

**Published:** 2025-03-21

**Authors:** Maya Faissal Alhomieed, Lara Osama Al Hartany, Marya Aref Alghorab, Arwa Alsharif, Ahlam Kaleemullah, Hanan Ismail Wasaya, Khlood Abdulaziz Alsubaie, Ayah Nabil Al Jehani, Amal Mohamed Kayali, Samera AlBasri

**Affiliations:** 1Department of Medicine and Surgery, Batterjee Medical College, Jeddah 21442, Saudi Arabia; 150011.maya@bmc.edu.sa (M.F.A.); 140044.lara@bmc.edu.sa (L.O.A.H.); 110010.marya@bmc.edu.sa (M.A.A.); 140300.ahlam@bmc.edu.sa (A.K.); 140204.hanan@bmc.edu.sa (H.I.W.); 140302.khlood@bmc.edu.sa (K.A.A.); 130055.ayah@bmc.edu.sa (A.N.A.J.); 130024.amal@bmc.edu.sa (A.M.K.); 2Obsetetrics and Gynecology Department, King Abdulaziz University Hospital, Faculty of Medicine, King Abdulaziz University, Jeddah 21589, Saudi Arabia; salbasri@kau.edu.sa

**Keywords:** pregnancy dermatoses, skin disorder, maternal health, intrahepatic cholestasis (ICP), pemphigoid gestationis (PG), atopic eruption of pregnancy (AEP), polymorphic eruption of pregnancy (PEP), dermatology in pregnancy

## Abstract

**Background**: Pregnancy induces hormonal, immunologic, and vascular changes that profoundly affect dermatologic health. This systematic review aimed to assess the impact of pregnancy on dermatological disorders in terms of disease incidence, severity, maternal-fetal outcomes, and optimal management strategies. **Methods**: A systematic search was performed in PubMed, MEDLINE, and Web of Science databases, following PRISMA guidelines. Studies evaluating pregnant women with dermatological disorders, pregnancy-related dermatoses, and pre-existing morbidities, were included. The collaboratively extracted data included patient demographics, disease severity, treatment approaches, and pregnancy outcomes. **Results**: A total of 8490 pregnant cases with dermatologic changes and conditions caused by pregnancy were studied. The dermatological conditions were divided into physiological changes, pregnancy-related exacerbation of pre-existing skin conditions, and pregnancy-specific dermatoses. Intrahepatic cholestasis of pregnancy and pemphigoid gestationis were associated with increased rates of adverse fetal outcomes in patients with specific dermatoses, including increased preterm birth and fetal distress rates. The atopic eruption of pregnancy and polymorphic eruption of pregnancy were highly relevant, but their effect on fetal health was minimal. The efficacy and safety of treatment modalities, including corticosteroids, antihistamines, and ursodeoxycholic acid, were variable. **Conclusions**: Pregnancy drastically affects dermatological health, but the nature of the impact depends on the condition. Optimal maternal and fetal outcomes rely on early diagnosis and individualized management strategies. More randomized controlled trials are required to develop standardized diagnostic and treatment guidelines to enhance the quality of dermatologic care during pregnancy.

## 1. Introduction

Pregnancy is a physiological state characterized by distinct immunological, hormonal, and vascular alterations that might affect dermatological health [1]. Up to 90% of pregnant women experience changes in their skin, ranging from benign physiological changes to exacerbations of pre-existing dermatological conditions and the appearance of pregnancy dermatoses [2]. The dermatological alterations are associated with fluctuations in estrogen, progesterone, and other hormones pertinent to pregnancy, as well as immune system modifications that promote fetal development [3].

Pregnancy dermatoses can be categorized into three primary groups: normal physiological skin alterations, worsening of pre-existing skin conditions due to pregnancy, and pregnancy-specific dermatoses [4]. Common physiological changes, typically reversing postpartum, include hyperpigmentation (melasma and linea nigra), and striae gravidarum [5]. Pre-existing dermatological diseases, including atopic dermatitis, psoriasis, and autoimmune bullous illnesses, might be dramatically impacted. Pregnancy-specific dermatoses, including pemphigoid gestationis (PG), polymorphic eruption of pregnancy (PEP), atopic eruption of pregnancy (AEP), and intrahepatic cholestasis of pregnancy (ICP), exhibit distinct diagnostic and treatment challenges and might lead to detrimental maternal and fetal consequences [6].

The start timing, intensity, and progression of dermatological illnesses during pregnancy are influenced by various factors, including maternal age, parity, genetic susceptibility, and immunological tolerance mechanisms. Some conditions resolve spontaneously after delivery, while others necessitate tailored intervention to avert unnecessary complications [7]. The clinical management of dermatological diseases in pregnant individuals must be cautious, prioritizing optimal results for the mother and fetus, particularly in treatment selection, to minimize teratogenic risk [8].

This systematic review aimed to assess the influence of pregnancy on several dermatological conditions, including their incidence, severity, progression, and implications for mother and fetal health. Our objective was to evaluate the outcomes of dermatological disorders and to determine if pregnancy exacerbates, ameliorates, or has no meaningful impact on these conditions. Despite existing literature on skin alterations and their therapy during pregnancy, a consensus on the management of many dermatologic diseases during this period appears to be lacking. Advocates of early intervention emphasize the advantages of symptom management and problem prevention, but others propose a more cautious strategy due to the restricted treatment options deemed safe during pregnancy. This study aimed to provide evidence-based dermatological treatment recommendations to assist doctors in achieving an optimal balance between safety and efficacy for mothers and fetuses.

## 2. Materials and Methods

### 2.1. Search Strategy

Our systematic review was registered with PROSPERO (CRD42025635819, https://www.crd.york.ac.uk/prospero/, accessed on 8 January 2025) and conducted according to the Preferred Reporting Items for Systematic Reviews and Meta-Analyses (PRISMA) guidelines (Appendix A and Appendix B) [9]. In PubMed, MEDLINE, and Web of Science databases without a time limit, a thorough electronic search using a predetermined search strategy developed by 2 authors (S.A. and A.A.) and approved by the remainder of the study team. For identifying relevant studies that acquired the effects of pregnancy on dermatological disorders, a comprehensive search of the PubMed database was performed using Medical Subject Headings (MeSH) terms “Pregnancy Dermatoses”, “Gestational Skin Disorders”, or “Dermatology in Pregnancy” in combination with MeSH terms “Hormonal Influence”, “Pregnancy-Related Skin Changes”, or “Autoimmune Skin Disorders in Pregnancy” to identify research that examined the effects of pregnancy on dermatological disorders as well as “Complications” or “Maternal-Fetal Outcomes”. The references of the selected articles were checked to find any articles that might have been missed.

### 2.2. Study Selection

#### 2.2.1. Inclusion Criteria

Studies that evaluated the impact of pregnancy on dermatological disorders, including their severity, progression, and outcomes for both the mother and the fetus, were included in this systematic review. Studies assessing autoimmune dermatoses like pemphigus vulgaris, dermatomyositis, systemic lupus erythematosus (SLE), and PG were considered. Additionally, there are immune-mediated chronic inflammatory skin disorders such as atopic dermatitis (AD) and psoriasis. Additionally, it covered immune dysregulation-related pregnancy-specific diseases such as ICP and PEP, which entail hormonal and inflammatory interactions that may impair immunological function. RCTs, quasi-experimental, cohort, case-control, systematic review, case report, and observational studies published in English were all included in this study. Studies from reputable sources (PubMed, MEDLINE, and Web of Science) as well as peer-reviewed publications were included.

#### 2.2.2. Exclusion Criteria

We excluded studies that did not evaluate the effect of pregnancy on dermatological disorders, including studies focusing on dermatological conditions unrelated to pregnancy, animal investigations, in vitro assays, and research published in non-English languages. Studies with incomplete or missing data on key variables necessary for analysis were also excluded.

#### 2.2.3. Screening and Data Extraction

The remaining results were imported into Rayyan [(https://www.rayyan.ai/), accessed 8 November 2024] by five authors (M.F.A., L.O.A., M.A.A. and A.K.) and screened by title and abstracts for relevance to the aims of the study [10]. All the studies were checked at least by two authors to avoid discrepancy and bias. After this initial screening, three authors (K.A.A., A.A., and A.M.K.) conducted a full-text review of publications that were part of the first screening to make a final decision on inclusion or exclusion according to the eligibility criteria adopted. Any discrepancies during selection were resolved through structured discussion and consensus meetings that included an additional reviewer (S.A.) and senior researchers, as required.

An Excel sheet was done by A.A., A.M.K., and S.A. to extract data systematically on the following study characteristics title, author name, country, year of publication, journal name, study design, level of evidence, sample size, reported dermatological complications, maternal-fetal outcomes, disease severity, and management strategies.

#### 2.2.4. Quality Assessment and Bias Evaluation

The included studies’ quality of evidence and risk of bias were evaluated using the Grading of Recommendations Assessment, Development, and Evaluation (GRADE) approach [11]. This comprehensive assessment was used to determine the degree of evidence in all research, resulting in an overall quality score that showed a risk of bias. Both prospective and retrospective cohort studies were evaluated for bias using the Newcastle-Ottawa Scale (Appendix C) [12]. The updated Cochrane Risk of Bias tool for randomized trials (RoB 2; Appendix E) was used to assess the bias risk of RCTs [13]. The MINORS tool (Appendix D) was used to evaluate nonrandomized studies [14]. These assessments increase the credibility of the results presented by educating the reader about the caliber of the included studies and possible sources of bias.

### 2.3. Data Synthesis

Due to high inconsistency in data formats and high heterogeneity, a meta-analysis was not feasible; this substantially inhibited the ability to synthesize the findings and arrive at firm and concrete conclusions. This heterogeneity was due to several factors, including differences in study design (e.g., RCTs, cohort studies, and observational studies), as well as differences in methodologies and tools to assess outcomes. Differences in patient populations (e.g., differences in gestational age, differences in underlying dermatological conditions, differences in baseline characteristics, etc.) further contributed to study heterogeneity. In addition, outcome measures varied widely, with varied primary endpoints reported across studies, including symptom severity, complication rates, treatment response, and recurrence. Differences in the definition and reporting of clinical outcomes prevented direct comparisons between studies. The treatment modalities used also varied substantially, from topical therapies to systemic treatments with varied dosing regimens and follow-up timelines that made it difficult to assess the efficacy of varied treatments in studies.

Such methodological and clinical differences precluded statistical pooling and limited the generalizability of findings. So, although some individual studies offered useful insights, the weight of the evidence overall was compromised. The findings highlight the significant heterogeneity between studies and that future research should utilize standardized study designs, standardized outcome measures, and harmonized reporting standards, which, in combination, would improve comparability and ensure that clinical decisions regarding the management of dermatological disorders in pregnancy are evidence-based.

## 3. Results

### 3.1. Study Selection and Characteristics

In total, the search identified 6260 unique records from PubMed (*n* = 2436), MEDLINE (*n =* 2176), and Web of Science (*n* = 1648). After the exclusion of field drifts (duplicates and irrelevant records), 1348 studies were screened. After a full-text evaluation, 76 studies stained the inclusion criteria for this systematic review of pregnancy-related dermatological disorders. The studies included in this systematic review were RCTs, cohort studies, case-control studies and case reports. The majority were observational (cohort and case-control) followed by RCTs. Among the studies, there was diversity in geography (North America, Europe, Asia, and Africa), though there was a lack of studies conducted in low-income regions. This introduces concerns about generalizability, as ethnicity-specific predisposition, access to healthcare, and environmental factors may affect the prevalence, severity, and management of disease. The age of the pregnant women ranges across the included studies from 18 to 45 years, covering adolescents, reproductive-age women, and those with advanced maternal age.

Meta-analysis according to Meta-analysis of Observational Studies in Epidemiology guidelines (MOOSE) was not performed given the heterogeneity of study designs and outcome measures, however, the systematic review was structured on PRISMA guidelines for transparent reporting of review findings. The PRISMA flowchart in Figure 1 summarizes the study selection process. Table 1 summarizes the main characteristics per study, including the sample size, country of origin, and the level of evidence.

### 3.2. Dermatological Disorders in Pregnancy

The included studies described pregnancy-related dermatoses, classified as pregnancy-specific dermatoses [37], which are PG [38], ICP [39], AEP [40,41], PEP [42], and pustular psoriasis of pregnancy (PPP) [43,44]. And pre-existing dermatological conditions that may worsen in pregnancy are AD [45], psoriasis [46], and lupus erythematosus (LE) [47].

#### 3.2.1. Polymorphic Eruption of Pregnancy

PEP is one of the most common pregnancy-specific dermatoses, characteristically occurring in third trimester and commoner in primigravida [48]. Intense pruritus, urticarial papules, and plaques in abdominal striae were the hallmark symptoms in the studies [49]. PEP is not fetotoxic, but severe pruritus can lead to sleep disturbances and emotional distress requiring symptomatic treatment with topical corticosteroids, antihistamines, and emollients [50,51] (Table 2).

#### 3.2.2. Atopic Dermatitis and Biologic Therapy in Pregnancy

AD typically exacerbates in pregnancy because of hormonal changes and immune modulation towards a Th2 profile. In severe cases systemic therapy may be needed, including biologic agents such as dupilumab. However, the evidence for the safety of biologics in pregnancy is limited. Table 3 summarizes the risk factors for worsening AD, including high IgE levels, genetic predisposition, and pre-existing atopy.

#### 3.2.3. Lupus Erythematosus and Pregnancy

LE creates distinct challenges in pregnancy because flares can cause severe maternal and fetal morbidity. The study concluded that SLE is linked with an elevated risk of preterm birth, preeclampsia, and neonatal lupus syndrome. Disease activity is associated with estrogen-driven immune dysregulation, and some cases worsen during the third trimester. Systemic corticosteroids and hydroxychloroquine continue to be the principal agents, but immunosuppression may be needed for refractory cases.

### 3.3. Maternal and Fetal Complications

Maternal and fetal complications related to pregnancy dermatoses are summarized in Table 2. PG is associated with small-for-gestational-age infants and preterm delivery. ICP is strongly linked to stillbirth and fetal distress, leading to early delivery in severe instances. AEP and PEP causes maternal discomfort without direct fetal risks.

Dermatological disorders in high-risk pregnancies (multiple gestations, advanced maternal age, etc.) were associated with more cutaneous complications and greater maternal-fetal morbidity.

### 3.4. Treatment Approaches and Variability in Effectiveness

There were significant differences in treatment strategies between studies (Table 2 and Figure 2), and systemic corticosteroids, antihistamines, and UDCA were the most used therapies. However, the effectiveness and safety profiles varied based on differences in the type of study (RCTs versus observational studies), dosage, and duration of corticosteroid therapy, when treatment is initiated gestational age and mapping comorbid conditions that affect treatment response.

Systemic corticosteroids (e.g., prednisone, methylprednisolone) were also frequently used for PG and pustular psoriasis of pregnancy but were associated with gestational diabetes, hypertension, and intrauterine growth restriction in some studies. Despite those risks, they are the first-line treatment for severe cases.

As for ICP, UDCA is generally recommended for its ability to reduce bile acid levels to help alleviate maternal pruritus. Studies produced divergent findings regarding its safety in the prevention of stillbirth, leading to the argument of performing more RCTs to adjust treatment measures.

Limited data exist regarding biologic therapies (e.g., TNF-α inhibitors, dupilumab) and immunosuppressants (e.g., azathioprine, cyclosporine) during pregnancy. Patients presenting with more severe, refractory AD, psoriasis, or lupus need careful consideration in terms of the risk/benefit ratio, according to current clinical evidence.

## 4. Discussion

This systematic review focused on pregnancy and dermatological diseases, emphasizing that physiological, hormonal, and immunological changes modify both pre-existing and pregnancy-specific dermatoses. This is because pregnancy causes benign physiological changes (e.g., melasma, linea nigra), or it exacerbate pre-existing dermatological conditions, and therefore requires individual clinical management.

Our study confirmed the existing data that pigmentation changes (melasma and linea nigra) are associated with major physiological skin changes during pregnancy, with a prevalence of up to 90% [61]. These alterations are primarily due to higher levels of estrogen, progesterone, and melanocyte-stimulating hormone, leading to increased melanogenesis and altered skin pigmentation [62]. Although these changes are mostly benign, they can be cosmetically bothersome to patients who require reassurance and management strategies for postpartum care. Among pregnancy-specific dermatoses, AEP is the most common chronic disorder, accounting for 48.8% of cases, most occurring in the first or second trimester, followed by PEP, PG, and ICP [63]. The accumulation of bile acids in ICP leads to significant fetal risks, such as premature birth, fetal distress, and stillbirth. This malady presented mainly with pruritus [64]. These data underscore the importance of careful surveillance and early treatment with ursodeoxycholic acid to reduce the risk of adverse perinatal outcomes [65].

PV is an autoimmune-mediated blistering disorder that increases the risk of fetal growth restriction and preterm birth [66]. Corticosteroids are effective in controlling autoimmune-mediated blistering, but studies have suggested that long-term systemic corticosteroid exposure possibly increases the risk of gestational diabetes, hypertension, intrauterine growth restriction, and preterm birth [67]. Despite these issues, systemic corticosteroids are still the first-line therapy due to their efficacy for controlling inflammation and preventing disease progression. Nonetheless, clearly designed RCTs are required to assess its long-term effects on maternal health, fetal entailment, and the risk-benefit balance of various dosages and treatment periods [68]. PEP is a benign yet troublesome condition that occurs later in pregnancy, primarily in primigravida, and is treated with topical corticosteroids, antihistamines, and emollients [69].

The role of pre-existing dermatological conditions in pregnancy was another area of exploration. Our overview suggests that pregnancy induced Th2 dominance may worsen AD by prompting severe flares. Topical corticosteroids and topical emollients are commonly used but for severe cases systemic therapies, including biologics like dupilumab, may be required. Nonetheless, the lack of data on biologic safety in pregnancy highlights the need for additional study [70]. It also suggests that psoriasis classically improves in pregnancy, possibly due to immune modulation that favors Th2 cytokines, while SLE and pemphigus are exacerbated, conferring an increased risk of fetal complications, preterm delivery and maternal disease flares. LE is associated with higher rates of maternal and fetal morbidity, such as preeclampsia and neonatal lupus syndrome. Management involves hydroxychloroquine, corticosteroids, and immunosuppressants with careful monitoring [71].

Only a few studies dealt with the risk of malignant diseases during pregnancy, but some limited evidence suggested an elevated melanomas incidence. Hormonal changes may also affect melanoma progression, the precise relationship is still unclear. More studies are needed to assess pregnancy-associated malignancies regarding its effect on maternal-fetal outcomes [72].

Treatment options for pregnancy-associated dermatological disorders were heterogeneous among the studies, with systemic corticosteroids, antihistamines and ursodeoxycholic acid (UDCA) found to be the most prescribed treatments. However, their efficacy and safety profiles remain issues of debate [73]. Corticosteroids are effective agents for PG and pustular psoriasis of pregnancy, with potential for maternal metabolic complications and fetal growth restriction [74]. The use of UDCA is neither supported nor questioned in preventing stillbirth in ICP, necessitating larger and well-controlled RCTs [75]. Although biologics and immunosuppressants (TNF-α inhibitors, dupilumab, etc.) may be required for AD and/or psoriasis with significant severity, their long-term fetal safety is not well understood, and more investigation is needed for this population [76]. Standardized treatment protocols were not documented across studies which present the need for some multicenter RCTs, at least at a regional level, to establish baseline management guidelines that would allow dermatologists to make evidence-based clinical decisions when managing dermatological disorders during pregnancy.

### Limitations

The studies included had different sample sizes, study designs, and diagnostic criteria, which made the results difficult to compare. Many did not have standardized measures of outcomes, so assessing disease severity, progression, and treatment response was challenging. Many of the studies were performed in specialized dermatology or obstetric referral hospitals, which may have led to an overestimation of the prevalence of severe dermatological diseases and an underestimation of milder physiologic changes. Most of the studies were conducted in North America, Europe, and Asia, with few studies from low-resource contexts, which may limit the generalizability of study findings.

Moreover, many of the studies failed to control for pre-existing dermatologic conditions, making it difficult to isolate pregnancy as the only modifying factor in disease progress. Given the heterogeneity in the corticosteroid dosing regimens and in the use of UDCA, direct comparisons of therapeutic effectiveness were not possible. However, there is limited information concerning the long-term outcomes in mothers and neonates, specifically regarding the recurrence of pregnancy-associated dermatological disorders in subsequent pregnancies.

To fill these gaps, multicenter studies in the future should have systemic diagnostic criteria and long-term follow-up. Include factors related to socioeconomic and health care access will facilitate assessment of dermatological care gaps during pregnancy.

## 5. Conclusions

This systematic review highlighted the importance of early recognition and management of pregnancy-related dermatological diseases, as these might play a significant role in avoiding maternal and fetal adverse events. It also highlighted the need for an individualized management approach based on gestational age, and comorbidities to provide the best care for mother and child. Topical corticosteroids, antihistamines, and ursodeoxycholic acid continue to represent the mainstay treatment, however, more studies are warranted to evaluate the safety of systemic immunosuppressants and other promising treatments to establish evidence-based guidelines.

## Figures and Tables

**Figure 1 clinpract-15-00068-f001:**
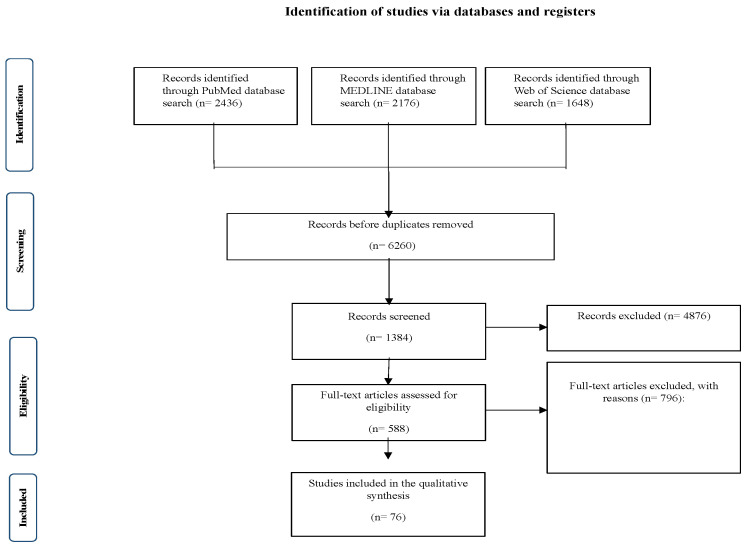
The phases of the study’s selection procedure in a comprehensive PRISMA chart utilized for the systematic review.

**Figure 2 clinpract-15-00068-f002:**
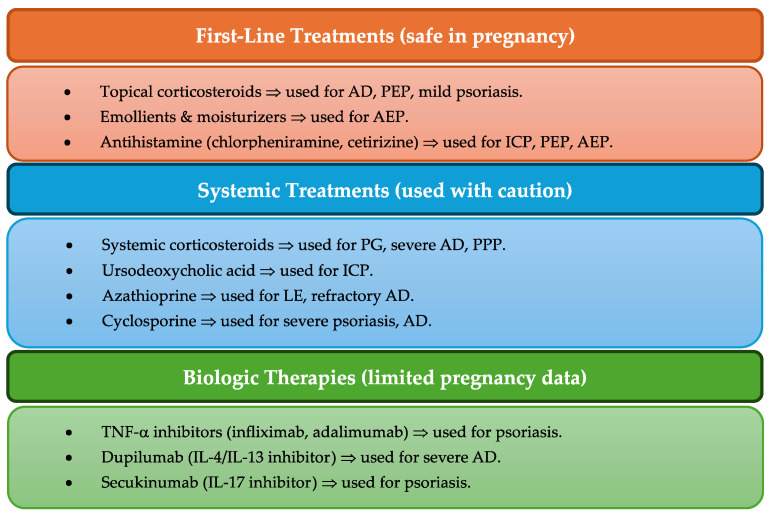
Overview of treatment approaches for pregnancy-related dermatological disorders.

**Table 1 clinpract-15-00068-t001:** Characteristics and outcomes of studies investigating the effect of pregnancy on dermatological disorders.

Authors	Country	Study Design	Sample Size	Summary	Level of Evidence
Thappa, et al. [15]	India	RCTs	607	Pigmentary changes were the most frequent specific dermatosis, while candidal vaginitis was the most common dermatosis affected by pregnancy.	IV
Szczęch, et al. [16]	Poland	RCTs	292	Pruritus is a common physiological or indicative underlying condition in pregnancy. Proper diagnosis and management are needed to prevent adverse maternal and fetal outcomes.	IV
Wong JH. [17]	USA	Retrospective Cohort Study	66	The study found that pregnancy does not negatively affect the prognosis of stage I melanoma.	III
Cobo, et al. [18]	Brazil	Clinical Study	7	Pregnant women with PG who were positive for PG factor and BP180 reactivity. They had a good outcome with no skin lesions and normal birth parameters.	IV
Nowecki, et al. [19]	Poland	Case Report	5	The study found that there is no clear evidence suggesting that pregnancy worsens melanoma prognosis.	V
Bushkell, et al. [20]	South Africa	Case Report	1	A patient with PG and a history of repeated abortion and stillbirths, contradicting the modern concept of a favorable fetal prognosis.	V
Bercovitch L [21]	USA	Case Report	1	A pregnant woman with pustular PG presented with no vesicular or bullous lesions.	V
Yaliwal, et al. [22]	India	Case Report	1	The study found that prurigo of pregnancy is a benign, pruritic skin disorder occurring during pregnancy. Management involves symptomatic relief, with no significant maternal or fetal complications.	V
Aronson, et al. [23]	USA	Clinical Study	57	The study categorized the PG into three types based on the clinical presentations: urticarial papules and plaques, no urticarial erythema/papules/vesicles, and mixed forms.	IV
Yancey, et al. [24]	Maryland	Prospective Study	25	Pruritus was the major symptom of PUPPP, but excoriations were rare in the third trimester of pregnancy.	II
Rudolph, et al. [25]	Austria	Multicenter Retrospective Study	181	PEP mainly affected primigravidae; it began as pruritic urticarial papules and plaques, evolving into polymorphic features in over 50% of cases.	III
Kroumpouzos, et al. [26]	USA	Case Report	1	A rare case of pruritic folliculitis of pregnancy, a dermatosis that resolved spontaneously postpartum.	V
Roger D [27]	France	Prospective Study	3192	The study highlighted the higher-than-expected incidence of PG (1 in 1700) and the need to consider pruritus gravidarum (1 in 145) in pregnancy-related itching.	II
Ambros-Rudolph, et al. [28]	Austria	Multicenter Retrospective Study	505	The study reclassified pregnancy dermatoses into PG, PEP, AEP, and ICP groups. AEP started early, while PEP, PG, and ICP presented in late pregnancy.	III
Chávez, et al. [29]	Mexico	Case Report	1	The study indicated that the SARS-CoV-2 infection should be considered in differential diagnoses of erythematous maculopapular rashes during pregnancy.	V
Mehedintu, et al. [30]	Romania	Case Report	1	The study highlighted the need for more awareness of PUPPP in pregnancy.	V
Liu, et al. [31]	China	Case Report	1	The study shows that pregnancy-related or hormonal factors may have contributed to the development of nodular vulgar lesions.	V
Kannambal, et al. [32]	India	Cross-sectional Study	500	The study highlighted the importance of differentiating benign changes from pathological conditions.	IV
Muzaffar, et al. [33]	Pakistan	Observational Study	140	The most common physiologic skin changes are pigmentation (90.7%) and striae (77.1%).	III
Fernandes, et al. [34]	Brazil	Cross-sectional Study	905	No significant differences in skin changes were observed between low-risk and high-risk pregnancies.	IV
Rathore, et al. [35]	India	Prospective Study	2000	The study found physiological cutaneous changes in 87.55% of cases, with pigmentary changes the most common (85.9%), followed by connective tissue changes (64.8%).	II
Aiholli, et al. [36]	India	Case Report	1	The study found that neonatal pemphigus vulgaris is a rare condition caused by transplacental transfer of maternal autoantibodies against desmogleins 1 and 3.	V

Abbreviations: RCTs: randomized controlled trials; BP180: bullous pemphigoid 180; PUPPP: pruritic urticarial papules and plaques of pregnancy; PEP: polymorphic eruption of pregnancy; PG: pemphigoid gestationis; AEP: atopic eruption of pregnancy; ICP: intrahepatic cholestasis of pregnancy; PPP: pustular psoriasis of pregnancy.

**Table 2 clinpract-15-00068-t002:** Patient-Reported Outcomes and Complications of Pregnancy-Related Dermatological Disorders.

Authors	Dermatosis	Primary Symptoms	Maternal Complications	Fetal Complications	Common Treatments
Sävervall, et al. [51]	Pemphigoid gestationis	Pruritic blisters, erythematous plaques	Increased risk of postpartum flares, association with autoimmune diseases	Small-for-gestational-age infants, preterm birth	Systemic corticosteroids, antihistamines
Kondrackiene, et al. [52]	Intrahepatic cholestasis of pregnancy	Intense pruritus (especially on palms and soles), jaundice	Increased bile acid levels, liver dysfunction	Preterm birth, stillbirth, fetal distress	Ursodeoxycholic acid, early delivery if bile acids are high
Massone, et al. [53]	Atopic eruption of pregnancy	Eczematous plaques, excoriations, dryness	Sleep disturbances, worsening of pre-existing atopy	None	Emollients, topical corticosteroids, antihistamines
Charles-Holmes R [54]	Polymorphic eruption of pregnancy	Urticarial papules, pruritic rash in abdominal striae	Severe itching, sleep disruption	None	Topical corticosteroids, antihistamines, emollients
Breier-Maly, et al. [55]	Pustular psoriasis of pregnancy	Pustular lesions, systemic symptoms (fever, malaise)	Risk of secondary infections, severe systemic inflammation	Increased risk of fetal growth restriction, preterm birth, stillbirth	Systemic corticosteroids, immunosuppressive therapy in severe cases

**Table 3 clinpract-15-00068-t003:** Association Between Pregnancy-Related Dermatological Disorders and Risk Factors.

Authors	Dermatosis	Hormonal Influence	Genetic Predisposition	Gestational Age at Onset	Multiparity vs. Primiparity	Pre-Existing Skin Condition
Shornick, et al. [56]	Pemphigoid gestationis	High estrogen/progesterone fluctuations	HLA-DR3, HLA-DR4 association	Second to third trimester	More common in multiparous women	Associated with other autoimmune diseases (e.g., Graves’ disease)
Wasmuth, et al. [57]	Intrahepatic cholestasis of pregnancy	Increased estrogen affects bile acid clearance	Family history of ICP, genetic mutations (ABCB4, ABCB11)	Third trimester	More common in multiparous women	Not strongly linked to pre-existing skin conditions
Stefaniak, et al. [58]	Atopic eruption of pregnancy	Elevated IgE, Th2 immune shift	Strong familial history of atopy (asthma, eczema)	First trimester	More common in primigravidas	Pre-existing eczema and atopic dermatitis increase the risk
Zejnullahu, et al. [59]	Polymorphic eruption of pregnancy	Skin stretching, hormonal factors uncertain	No known genetic link	Third trimester	More common in primigravidas	No strong association with pre-existing skin conditions
Liu, et al. [60]	Pustular Psoriasis of Pregnancy	Dysregulated immune response, IL-17 involvement	Strong genetic link to psoriasis-associated loci (HLA-Cw6)	Third trimester	Occurs in both primiparous and multiparous women	Pre-existing psoriasis and inflammatory conditions increase the risk

## Data Availability

Not applicable due to the nature of the study.

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
