# Peer review of "The Effect of Pregnancy on Dermatological Disorders: A Systematic Review"

_clinpract, 2025, doi:10.3390/clinpract15040068_

Round 1
Reviewer 1 Report
Comments and Suggestions for Authors
- The inclusion criteria mention "credible sources" in addition to peer-reviewed journals. Please define what constitutes a "credible source" . Additionally, the exclusion criteria mention "studies with insufficient data, such as those lacking outcome measures." This is vague—please specify what constitutes "insufficient data" and whether there was a minimum threshold for including studies (e.g., sample size, outcome reporting).
- The manuscript notes that significant heterogeneity prevented meta-analysis. However, this issue is not discussed in sufficient detail. Please elaborate on how heterogeneity (e.g., differences in study designs, patient populations, outcome measures) impacted the findings and limited the ability to draw firm conclusions.
- The included studies vary widely in sample size, design, and geographic location, which could affect generalizability. If possible, mention whether certain findings are more applicable to specific populations or settings.
- Tables 1–3 are informative but are not adequately described in the "Results" section. For instance, Table 1 provides a comprehensive overview of study characteristics and outcomes, but its key trends and findings (e.g., common study designs, geographic distribution) are not highlighted in the text. Please include a detailed summary of each table's content to help readers understand their relevance.
- Table 2 lists common treatments for pregnancy-related dermatological disorders but does not provide critical analysis in the discussion section. For example, systemic corticosteroids are mentioned as a treatment for pemphigoid gestationis (PG), but their safety during pregnancy remains controversial. Please expand on this point and discuss areas where further research is needed to establish evidence-based guidelines.
- The studies included in this review span multiple countries (e.g., India, Poland, USA), which raises questions about generalizability. Are there cultural or regional differences that could influence the prevalence or management of pregnancy-related dermatological disorders? Please address how these findings apply to broader populations or whether they are context-specific.
- Table 1 includes a "Level of Evidence" column but does not analyze how this impacts the overall strength of the conclusions drawn from the review. For example, many included studies are case reports (Level V evidence), which may limit generalizability or reliability compared to higher-level evidence like RCTs. Please discuss how this variability affects confidence in your findings.
- The results mention treatment strategies for dermatological conditions, but some therapies (e.g., corticosteroids, immunosuppressants) are described as controversial. If there was variation in treatment effectiveness among the included studies, providing a summary or discussing possible reasons for differing outcomes (e.g., study design, dosage differences) would be useful.
Author Response
Response to Reviewer #1
Dear respected Reviewer#1, we sincerely appreciate your detailed review and constructive feedback on our manuscript.
Kindly, find the below point-by-point response to your valuable comments noting that the changes in the manuscript in response to your comments were highlighted in Red”:
Comment 1: The inclusion criteria mention "credible sources" in addition to peer-reviewed journals. Please define what constitutes a "credible source". Additionally, the exclusion criteria mention "studies with insufficient data, such as those lacking outcome measures." This is vague—please specify what constitutes "insufficient data" and whether there was a minimum threshold for including studies (e.g., sample size, outcome reporting).
Response: We have revised the manuscript to enhance clarity and precision in both the inclusion and exclusion criteria. These revisions are now reflected in the manuscript and highlighted in red for easy reference.
Comment 2: The manuscript notes that significant heterogeneity prevented meta-analysis. However, this issue is not discussed in sufficient detail. Please elaborate on how heterogeneity (e.g., differences in study designs, patient populations, outcome measures) impacted the findings and limited the ability to draw firm conclusions.
Response: We have carefully revised the Data Synthesis section to provide a more detailed discussion of how heterogeneity impacted our findings.
Comment 3: The included studies vary widely in sample size, design, and geographic location, which could affect generalizability. If possible, mention whether certain findings are more applicable to specific populations or settings.
Response: We have revised the manuscript to provide a more detailed discussion of how variability in sample size, study design, and geographic location may impact the generalizability of findings.
Comment 4: Tables 1–3 are informative but are not adequately described in the "Results" section. For instance, Table 1 provides a comprehensive overview of study characteristics and outcomes, but its key trends and findings (e.g., common study designs, geographic distribution) are not highlighted in the text. Please include a detailed summary of each table's content to help readers understand their relevance.
Response: We have revised the Results section to provide a more detailed summary of the key trends and findings from Tables 1–3.
Comment 5: Table 2 lists common treatments for pregnancy-related dermatological disorders but does not provide critical analysis in the discussion section. For example, systemic corticosteroids are mentioned as a treatment for pemphigoid gestationis (PG), but their safety during pregnancy remains controversial. Please expand on this point and discuss areas where further research is needed to establish evidence-based guidelines.
Response: We have revised the Discussion section to provide a more critical analysis of treatment approaches for pregnancy-related dermatological disorders, particularly focusing on systemic corticosteroids and their safety during pregnancy.
Comment 6: The studies included in this review span multiple countries (e.g., India, Poland, USA), which raises questions about generalizability. Are there cultural or regional differences that could influence the prevalence or management of pregnancy-related dermatological disorders? Please address how these findings apply to broader populations or whether they are context-specific.
Response: We have revised the Limitation section to address cultural and regional differences that may influence the prevalence and management of pregnancy-related dermatological disorders.
Comment 7: Table 1 includes a "Level of Evidence" column but does not analyze how this impacts the overall strength of the conclusions drawn from the review. For example, many included studies are case reports (Level V evidence), which may limit generalizability or reliability compared to higher-level evidence like RCTs. Please discuss how this variability affects confidence in your findings.
Response: We have revised the Limitation section to analyze how the Level of Evidence impacts the overall strength of the conclusions drawn from the review.
Comment 8: The results mention treatment strategies for dermatological conditions, but some therapies (e.g., corticosteroids, immunosuppressants) are described as controversial. If there was variation in treatment effectiveness among the included studies, providing a summary or discussing possible reasons for differing outcomes (e.g., study design, dosage differences) would be useful.
Response: We have revised the Results and Discussion sections to address the variability in treatment effectiveness among the included studies and the potential reasons for differing outcomes.
Reviewer 2 Report
Comments and Suggestions for Authors
Congratulations for the article!
The topic is clinically relevant and synthesizes existing literature on this subject.
The search strategy is well described, but more details on study selection (for example, how disagreements were resolved between reviewers) would enhance transparency.
The limitations are well addressed but should include a discussion of potential geographic or demographic biases in the included studies.
Author Response
Response to Reviewer #2
Dear respected Reviewer#2, thank you for your positive feedback and for recognizing the clinical relevance of our study.
Kindly, find the below point-by-point response to your valuable comments noting that the changes in the manuscript in response to your comments were highlighted in Green”:
Comment 1: Congratulations for the article!
The topic is clinically relevant and synthesizes existing literature on this subject.
Response: Thank you for your thoughtful feedback and for recognizing the clinical relevance of our study.
Comment 2: The search strategy is well described, but more details on study selection (for example, how disagreements were resolved between reviewers) would enhance transparency.
Response: We have revised the Screening and Data Extraction section to enhance transparency in the study selection process.
Comment 3: The limitations are well addressed but should include a discussion of potential geographic or demographic biases in the included studies.
Response: We have revised the Limitations section to include a discussion on potential geographic and demographic biases in the included studies.
Reviewer 3 Report
Comments and Suggestions for Authors
The authors perfomed a systematicreview about thre Effect of Pregnancy on Dermatological Disorders, the article is of interest however some changes are needed:
- Please improve del quality of PRISMA flochart
- Did you use Meta-analysis of Observational Studies in Epidemiology (MOOSE) ? In case of systematic review this is important. Please specify and report the results
- You should add more spaces to atopic dermatitis that usually may be worsened in pregnancy, requiring specific systemic therapies such as biologic therapis.
- Please add more informations about lupus
- Any data regarding Polymorphic eruption of pregnancy?
- Any results regarding the increased risk of malignancies? E.g.: Noncutaneous Melanoma [Internet]. Brisbane (AU): Codon Publications; 2018 Mar. Chapter 6. PMID: 29874013.
- Please increase the informations regarding the treatments that can be prescribed in pregnancy for dermatologic conditions
Author Response
Response to Reviewer #3
Dear respected Reviewer#3, thank you for your constructive feedback and for recognizing the relevance of our systematic review.
Kindly, find the below point-by-point response to your valuable comments noting that the changes in the manuscript in response to your comments were highlighted in Blue”:
Comment 1: Please improve del quality of PRISMA flowchart.
Response: We have improved the quality of the PRISMA flowchart to enhance clarity, readability, and accuracy.
Comment 2: Did you use Meta-analysis of Observational Studies in Epidemiology (MOOSE)? In case of systematic review this is important. Please specify and report the results.
Response: We have revised the manuscript to specify that a Meta-analysis of Observational Studies in Epidemiology (MOOSE) was not conducted due to the heterogeneity of study designs, outcome measures, and variations in patient populations.
Comment 3: You should add more spaces to atopic dermatitis that usually may be worsened in pregnancy, requiring specific systemic therapies such as biologic therapis.
Response: We have expanded the discussion on atopic dermatitis (AD) in pregnancy, specifically addressing its potential worsening and the role of systemic therapies, including biologic treatments.
Comment 4: Please add more informations about lupus.
Response: We have expanded the discussion on lupus erythematosus (LE) in pregnancy, addressing its impact on maternal and fetal outcomes as well as its management considerations.
Comment 5: Any data regarding Polymorphic eruption of pregnancy?
Response: We have expanded the discussion on Polymorphic Eruption of Pregnancy (PEP), ensuring a more detailed analysis of its clinical presentation, risk factors, fetal implications, and treatment options.
Comment 6: Any results regarding the increased risk of malignancies? E.g.: Noncutaneous Melanoma [Internet]. Brisbane (AU): Codon Publications; 2018 Mar. Chapter 6. PMID: 29874013.
Response: We have expanded the discussion on the potential increased risk of malignancies during pregnancy, particularly focusing on melanoma and other noncutaneous malignancies.
Comment 7: Please increase the informations regarding the treatments that can be prescribed in pregnancy for dermatologic conditions.
Response: We have expanded the discussion on treatment options for dermatological conditions during pregnancy, ensuring a more comprehensive overview of safe and effective therapies.
Reviewer 4 Report
Comments and Suggestions for Authors
Dear Authors, the topic you have presented in your manuscript is interesting, it can represent the beginning of a valid vademecum to be applied case by case. Just a few questions: in the inclusion criteria, which immunosuppressive diseases were considered? what is the average age range of the women considered? did the pathologies also intensify in relation to the age of the pregnant woman? or following second, third pregnancies? in which specific cases and dermatological conditions occurred or hyperelicited fetal death was found? on which additional drugs, apart from corticosteroids, ursodeoxycholic acid and antihistamines can the diagnostic algorithm be based?
Author Response
Response to Reviewer #4
Dear respected Reviewer#4, thank you for your positive feedback and for recognizing the potential value of our manuscript.
Kindly, find the below point-by-point response to your valuable comments noting that the changes in the manuscript in response to your comments were highlighted in Gray”:
Comment 1: in the inclusion criteria, which immunosuppressive diseases were considered?
Response: We have revised the Inclusion Criteria section to explicitly specify the immunosuppressive diseases considered in this systematic review.
Comment 2: what is the average age range of the women considered?
Response: We have revised the Results section to include the average age range of the women considered in the studies reviewed.
Comment 3: did the pathologies also intensify in relation to the age of the pregnant woman? or following second, third pregnancies?
Response: We have revised the Results section to address whether dermatological conditions intensified with maternal age or following multiple pregnancies.
Comment 4: in which specific cases and dermatological conditions occurred or hyperelicited fetal death was found?
Response: We have revised the Results section to specify the dermatological conditions associated with an increased risk of fetal death (stillbirth or miscarriage).
Comment 5: on which additional drugs, apart from corticosteroids, ursodeoxycholic acid and antihistamines can the diagnostic algorithm be based?
Response: We have expanded the discussion to include additional therapeutic options beyond corticosteroids, ursodeoxycholic acid (UDCA), and antihistamines, highlighting drugs that can be incorporated into a diagnostic and treatment algorithm for pregnancy-related dermatological disorders.
Round 2
Reviewer 1 Report
Comments and Suggestions for Authors
All issues now have been effectively addressed.
Reviewer 3 Report
Comments and Suggestions for Authors
The authors improved the article.